# Imaging of the Temporomandibular Joint

**DOI:** 10.3390/diagnostics12041006

**Published:** 2022-04-16

**Authors:** Seyed Mohammad Gharavi, Yujie Qiao, Armaghan Faghihimehr, Josephina Vossen

**Affiliations:** Department of Radiology, West Hospital, VCU School of Medicine, Virginia Commonwealth University, 1200 East Broad Street, North Wing, Box 980470, Richmond, VA 23298-0470, USA; yujie.qiao@vcuhealth.org (Y.Q.); armaghan.faghihimehr@vcuhealth.org (A.F.)

**Keywords:** TMJ, temporomandibular joint, imaging, temporomandibular disorder, radiologist, internal derangement, arthritis, neoplasm

## Abstract

Temporomandibular disorder (TMD) is a common musculoskeletal condition that causes pain and disability for patients and imposes a high financial burden on the healthcare system. The most common cause of TMD is internal derangement, mainly secondary to articular disc displacement. Multiple other pathologies such as inflammatory arthritis, infection, and neoplasm can mimic internal derangement. MRI is the modality of choice for evaluation of the TMJ. Radiologists need to be familiar with the normal anatomy and function of the TMJ and MR imaging of the internal derangement and other less common pathologies of the TMJ.

## 1. Introduction

The temporomandibular joint (TMJ) plays a crucial role in mastication, jaw mobility, verbal, and emotional expression. The prevalence of chronic TMJ pain ranges from 5 to 31%, and the incidence of first-time pain is 4% per year [1,2]. Temporomandibular disorder (TMD) is a term used to describe several pathologies that involve the TMJ and surrounding bone and soft tissues. TMD is the second most common musculoskeletal condition after back pain to cause pain and disability with an annual cost of $4 billion in the United States [3]. The incidence of TMD peaks in the second to the fourth decade of life with a higher prevalence among women [4,5]. While internal derangement is the most common TMJ pathology, radiologists should not overlook other less common pathologies such as inflammatory arthritis, infection, trauma, and neoplasm.

We aim to review normal TMJ anatomy and function, most common TMDs, and their imaging presentations.

## 2. Normal Anatomy

Interpreting TMJ imaging requires an understanding of the normal anatomy of the joint. TMJ is a synovial joint between the glenoid fossa of the temporal bone and the mandibular condyle. The central anatomic structure of the TMJ is the articular disc or meniscus. The disc is an oval-shaped fibrocartilaginous structure composed of anterior and posterior articular bands and a thinner center, called the intermediate zone. The intermediate zone gives the disc a biconcave appearance on the sagittal view. The posterior band is usually thicker than the anterior band, and both bands are wider in the transverse dimension than in the anteroposterior dimension. The retrodiscal tissue or bilaminar zone is a rich neurovascular tissue that serves as a posterior disc attachment, blending the disc with the joint capsule and temporal bone. The lateral aspect of the disc connects with the joint capsule and inserts into the condylar neck. Anteriorly, the attachments of the disc are variable and called the “disc-capsular complex”. The lateral pterygoid muscle fibers and tendons attach the anterior band of the disc to this complex. The position of the disc is evaluated by the location of the posterior band and intermediate zone in relation to the mandibular condyle. In the closed-mouth position, the posterior band should lie near the 12 o’clock position on the sagittal projection. The intermediate zone should be located between the condyle and the temporal bone. The medial and lateral corners of the disc align with the condylar borders and do not bulge medially or laterally [6].

TMJ has two compartments that function as two small joints within a same capsule. This allows for a greater range of motion with respect to the joint’s size. The superior compartment separates the glenoid fossa of the temporal bone from the disc, while the inferior compartment separates the disc from the mandibular condyle. In the initial phase of mouth opening, the condyle rotates in the lower joint compartment. Subsequently, it translates anteriorly in the upper compartment [6]. The lateral pterygoid muscle contributes to jaw opening, and the medial pterygoid, masseter, and temporalis muscles facilitate jaw closure. As the condyle translates anteriorly, the disc should move between the condyle and the articular eminence. A normal disc does not move in the coronal plane during mouth opening (Figure 1 and Figure 2).

## 3. Imaging

TMJ radiographic examinations include transcranial (oblique lateral view), trans maxillary (modified AP view), trans pharyngeal (oblique lateral view), and submental vertex radiographs. Cross-sectional imaging is usually indicated in cases where malocclusion or intra-articular abnormalities are suspected.

With its ability to evaluate the bones, CT is the mainstay imaging modality in the setting of trauma. CT is also particularly valuable in assessing surgical reconstruction, detecting calcified loose bodies, and in some cases of inflammatory, infectious, and neoplastic disease. Cone-beam computed tomography (CBCT) has shown comparable osseous detail to CT, with the advantage of decreased radiation dose.

MRI is the best modality for the evaluation of intra-articular processes. Given the high MRI contrast resolution of the soft tissues, it is currently the gold standard for diagnosing disc disorders. The standard MR imaging protocol consists of oblique sagittal and coronal proton density-weighted (PDWI) sequences in closed- and open-mouth positions. Images are acquired perpendicular or parallel to the long axis of the mandibular condyle to optimize the visualization of the disc and the osseous structures. In addition, T2WI is useful for detecting degenerative periarticular changes and the presence of a joint effusion. Gadolinium contrast is not used routinely but would be indicated in cases where infection, inflammatory arthropathy, or neoplasm is suspected.

Cine gradient echo (GRE) images depict disc movement and can be used to evaluate condylar translation. Newer dynamic techniques, such as a half-Fourier acquired single-shot turbo spin-echo (HASTE) or balanced steady-state free precession sequence (SSFP), can contribute to the evaluation of the disc movement. Due to higher signal-to-noise ratios, 3 T MRI magnets have the advantage of depicting improved anatomic and pathologic details of the TMJ compared with 1.5 T.

Ultrasound can be considered a non-invasive, readily available, and less expensive technique for the evaluation of the TMJ. The sonographer should perform a high-resolution ultrasound (12–20 MHz high-frequency linear transducer) by placing the probe perpendicular to the zygomatic arch and parallel to the mandibular condyle. The operator should acquire transverse and longitudinal images in closed-mouth and open-mouth positions. Adjusting and tilting the probe during the exam will help to optimize the visualization of the disc, condylar changes, and joint effusion [7,8]. Ultrasound evaluation is limited when evaluating deeper structures and/or any osseous abnormalities. On ultrasound, the normal articular disc appears as a hypoechoic, inverted c-shape structure, situated superior to the hyperechoic condylar cortex (Figure 3).

## 4. Pathology

### 4.1. Internal Derangement

The most common TMD is internal derangement (ID), which implies a mechanical interference of the smooth joint movement. ID peaks in the second to the fourth decade of life with a female-to-male ratio of 3:1 and most commonly presents with jaw pain, clicking, or locking [9,10]. The most common cause of ID is the displacement of the articular disc.

MRI is the standard imaging modality for assessing ID in TMJ [11]. Several key features should be evaluated while interpreting the MRI of TMJ derangement, including the position and morphology of the disc, the morphology and signal of the mandibular condyle, the presence of joint effusion, and condylar translation with dynamic imaging.

In a closed mouth position on an oblique sagittal plane, the posterior band of the disc should lie superior to the condylar head at the 12 o’clock position. The disc is anteriorly displaced if the posterior band lies more than 30 degrees anterior to the vertical line through the condylar head, at around 10 to 11 o’clock [12]. The condyle and disc both translate anteriorly as the mouth opens, and the disc stays between the articular eminence and mandibular condyle.

In the earliest stage of internal derangement, the disc has a normal biconcave morphology but is anteriorly displaced in the closed mouth position (Figure 4). However, the disc returns to the normal anatomical position or recaptures as the condyle translates anteriorly during mouth opening. In the intermediate stage, the disc still has a normal morphology, is displaced in a closed-mouth position, and does not recapture with mouth opening. In the later stages, the disc is chronically displaced and has an abnormal morphology, e.g., it is perforated or the posterior attachment to the bilaminar zone is disrupted. Imaging features of degenerative joint disease such as flattening of the condyle, osteophytes, joint effusion, or the abnormal T1 and T2 signal of the condyle can also be seen in more advanced stages [9,13,14,15]. A commonly used classification of TMJ disc displacement using clinical and imaging findings was described by Wilkes [16].

Ultrasound can be used as an alternative modality for evaluating ID, particularly for disc displacement and joint effusion. However, this modality is considered less useful in detecting bone abnormalities in condyle [8,17].

### 4.2. Osteoarthritis

Osteoarthritis (OA) is related to the breakdown of joint cartilage and underlying subchondral bone. Primary OA is more common in older patients. On the other hand, secondary OA can be seen in younger patients with ID, prior trauma, or other TMJ arthropathies [18]. The commonly seen imaging findings on CT or MRI are flattening of the condyles, osteophytes, bone erosions, joint space narrowing, subchondral sclerosis, and marrow signal abnormality [19]. Signs and symptoms of OA in TMJ are pain, movement limitations, and crepitus.

### 4.3. Avascular Necrosis

Avascular necrosis is the result of impaired blood flow to the condyle. It can be seen in the context of different etiologies such as trauma, sickle cell anemia, or systemic lupus erythematosus. On CT, the condyle of the mandible is deformed with significant subchondral sclerosis. On MR, the marrow is dark on T1WI and shows a mixed signal on T2WI [20].

### 4.4. Idiopathic Condylar Resorption

Idiopathic condylar resorption or “cheerleader syndrome” is a poorly understood disease that is most commonly occurs in adolescents and young women. This aggressive form of degenerative disease of the TMJ has been attributed to an exaggerated response to minor traumatic injury induced by excess estrogen receptors [9,21,22]. Imaging demonstrates loss of condylar bone mass and flattening of the anterior or superior aspect of the condyle (Figure 5).

### 4.5. Inflammatory Arthropathies

#### 4.5.1. Rheumatoid Arthritis

Rheumatoid arthritis (RA) is the most common inflammatory arthritis in adults that can affect the TMJ [23]. TMJ symptoms are seen in approximately 50% of patients with RA. However, TMJ involvement is not a classic finding of RA [24]. For most patients, the diagnosis of RA is made before the development of TMJ symptoms. The most common symptoms are pain, joint swelling, and limited jaw motion [25]. More severe involvement can lead to trismus, facial deformity, and chronic loss of function [26].

Contrast-enhanced MRI and CT are commonly used for the evaluation of inflammatory arthritis such as RA in the TMJ. Synovial proliferation, joint space narrowing, articular erosion, flattening of the condyle, disc deformity, shorter condylar height, and abnormal condylar motion are common imaging findings that are seen with RA [23,27,28,29].Unlike OA, in RA articular disc displacement occurs later in the course of disease and the disc can remain in a normal position despite substantial changes to the underlying condylar bone [30]. In patients with long-standing RA, disc displacement occurs more often due to the lack of bony support by the underlying osseous structures, not morphologic changes of the disc itself [30].

While no definite radiologic classification has been established to evaluate the severity of RA in the TMJ, Mohamed et al. showed that in patients with moderate to severe disease activity, the condyle has a smaller AP dimension. They also showed that disease activity had a statistically significant direct correlation with all osteoarthritic changes except for the glenoid and condylar erosions [29].

#### 4.5.2. Juvenile Idiopathic Arthritis

Juvenile idiopathic arthritis (JIA) is the most common rheumatologic disease of childhood and adolescents. The rate of TMJ involvement in JIA differs based on the subtypes, ranging from 40% to 70% [13,31]. Longer disease duration, early age at onset, polyarticular or systemic course, and lack of HLA-B27 are risk factors for TMJ involvement [28]. The early diagnosis and management of TMJ involvement in JIA patients are essential because a delay in diagnosis can damage the mandibular growth plate and compromise normal facial growth. Clinical diagnosis is complicated because the patient may be asymptomatic until a relatively late stage of the disease [30].

Contrast-enhanced MRI is the preferred imaging modality and shows both the acute findings and secondary degenerative arthritis. Synovial enhancement, joint effusion, and synovial thickening are the most common findings early in the disease process. As inflammation and joint destruction continue, chronic secondary arthritic changes such as pannus formation, bone erosions, and disc destruction become more prevalent, eventually leading to condylar flattening and deformity (Figure 6) [32].

Several scoring systems have been proposed for radiological grading of inflammation and damage of the temporomandibular joint in JIA patients based on the presence or degree of MRI findings such as bone marrow edema and enhancement, joint effusion, synovial thickening or enhancement, condylar flattening or erosion, and abnormal disc morphology [32,33].

#### 4.5.3. Calcium Pyrophosphate Deposition Disease

Calcium pyrophosphate deposition disease (CPPD) is a noninfectious inflammatory arthropathy characterized by crystal deposits in the articular and periarticular soft tissues [34]. Two primary forms of CPPD have been described, common and tumoral, with the tumoral type affecting the TMJ [35]. CPPD of the TMJ is rare overall, and only a few case reports have been published in the literature [36,37,38].

The definitive diagnosis of CPPD is made by joint aspiration and fluid analysis showing the presence of calcium pyrophosphate crystals, but the radiologic evaluation is critical for evaluating the extent of underlying osseous destruction, secondary osteoarthritis, the presence of joint fluid, and to exclude other etiologies. CT is the best imaging modality for this disease and generally shows a calcified mass in the joint with secondary destructive and degenerative changes [35]. Therefore, MRI is rarely used as the initial imaging modality if CPPD is suspected. However, MRI is still often performed since common symptoms of TMJ CPPD disease such as pain, joint swelling, and limited range of motion are also symptoms in other TMD, for which MRI is indicated as standard practice. Notable MRI features are periarticular T2 hypointense signals with heterogeneous enhancement, which can mimic more concerning diseases such as chondrosarcoma [35]. Treatment ranges from medical management to surgical debridement of the joint with or without possible resection of the involved condyle [37,39].

#### 4.5.4. Septic Arthritis

TMJ septic arthritis is rare but is associated with high morbidity and significant long-term disability [40]. The most common organism cultured is Staphylococcus aureus, which infects the joint either from direct inoculation or hematogenous spread [41].

Imaging findings of TMJ septic arthritis are similar to those of infections in other joints, namely synovial enhancement, joint effusion, surrounding soft tissue, and bone marrow edema. These findings are best visualized on MRI (Figure 7). Differentiation of septic arthritis from other inflammatory diseases is predominately based on the clinical presentation and the acuity of the symptoms. In septic arthritis, severe pain often occurs suddenly, with extreme tenderness on palpation, and is usually accompanied by other general symptoms such as malaise, fevers, or nausea/vomiting [41].

### 4.6. Trauma

Trauma to the jaw can cause a condylar fracture, glenoid fossa fracture, or TMJ dislocation. Condylar fracture accounts for 25–50% of mandibular fractures and is classified as the condylar head (intra- or extra-articular) or neck fracture [23].

In patients with a condylar fracture, the unopposed force of the lateral pterygoid muscle usually causes inferior and anteromedial dislocation of the condylar head and the lateral displacement and telescoping of the ramus (Figure 8). Multidetector CT is the modality of choice for evaluating facial and mandibular fractures in the acute setting. Most condylar fractures will show functionally favorable outcomes after closed reduction. However, traumatic dislocation of the disc or injury to the retrodiscal soft tissue can lead to joint ankylosis, a devastating complication. Studies have shown a correlation between the severity and pattern of condylar fractures on CT and risk of soft tissue injuries. However, MRI is the modality of choice to evaluate retrodiscal tissue injury and disc dislocation [42,43].

### 4.7. Tumor and Tumor-like Lesions

#### 4.7.1. Osteochondroma

Osteochondroma is a benign bone lesion that can arise from the mandibular condyle or glenoid fossa. It is considered to be the most common benign tumor of the TMJ [9]. On CT, it often appears as a pedunculated osseous mass, usually arising from the anterior surface of the condyle and at the insertion of the lateral pterygoid muscle (Figure 9). Continuity with the parent bone without cortical interruption is an essential characteristic feature. When small, it is challenging to differentiate osteochondroma from an osteophyte. Larger lesions can cause condylar displacement with associated pain or malocclusion [14,44]. Based on the growth pattern on CT, Chen et al. classified osteochondromas into two main types: type 1, which protrudes from the condyle and involves less than two-thirds of the surface of the condyle, and type 2, which is causing global expansion of the condyle [44].

MRI shows a predominantly low T1 signal exophytic mass with a T2-hyperintense cartilage cap. The size of the cartilage cap is directly related to the risk of malignant transformation to or harboring chondrosarcoma [45].

#### 4.7.2. Pigmented Villonodular Synovitis

Pigmented Villonodular Synovitis (PVNS) is a benign synovial proliferation process of uncertain origin, most commonly involving larger joints and uncommonly affecting the TMJ. Malignant PVNS is extremely rare but has been reported in the TMJ [46]. The CT depicts a soft tissue mass arising from the joint with areas of hyper attenuation which enhances after contrast injection. The adjacent bone might demonstrate erosions and sclerosis. MRI is the modality of choice for the diagnosis of PVNS, showing an intra-articular mass with areas of hemosiderin deposition depicted as low signal intensity on T1WI and T2WI and blooming artifact on GRE sequences [3].

#### 4.7.3. Synovial Chondromatosis

Synovial chondromatosis (SC) is a rare, benign synovial proliferative disease characterized by the growth of cartilaginous nodules in the synovium that eventually calcify and detach from the joint. As a result, SC usually presents with joint effusion and multiple loose bodies (Figure 10). Unlike PVNS, which never calcifies, the loose bodies in SC are typically calcified. Therefore, a CT can help differentiate the two entities [3,9].

#### 4.7.4. Chondrosarcoma

Chondrosarcoma is a malignant tumor that arises from embryogenic cartilaginous cells and is characterized by the production of a cartilaginous matrix. Chondrosarcoma can be primary—without pre-existent benign lesion—or secondary—arising from pre-existent benign lesions such as enchondroma or osteochondroma. Chondrosarcoma of the TMJ is extremely rare, with only 50 cases reported in the literature. The mean age of patients is 45.5 years with a female predominance (F:M = 1.4:1) [47]. The primary symptom of TMJ chondrosarcoma is preauricular swelling, followed by preauricular pain and trismus. Other findings, such as pain with mastication, obstruction of the external auditory canal, hearing loss, and limited jaw opening are commonly seen with all TMDs.

On CT, chondrosarcoma appears as a soft tissue mass with flocculent calcifications in the joint with or without bone erosion. A few cases with erosion of the glenoid fossa and intracranial invasion have been reported. On MRI, chondrosarcoma appears as an intermediate-to-low T1 and high T2 signal mass with foci of low T1 and T2 signal due to calcification. It shows heterogeneous enhancement on post-contrast images [47,48].

#### 4.7.5. Osteosarcoma

Osteosarcoma is a malignant bone tumor arising from the osteogenic mesenchymal matrix and producing osteoid, fibrous, cartilaginous, and osseous tissue. It usually involves the long bones but rarely involves the jaw and is called gnathic osteosarcoma. Jaw osteosarcoma is not as aggressive as osteosarcoma in the long bones, with the mean age of patients being 35 years–10 years younger than long bone osteosarcoma. In addition, there is a male predilection, with the male-to-female ratio of 2:1 [14,49]. Radiographically, osteosarcoma can have a lytic, sclerotic, or mixed appearance with malignant periosteal reaction. On MRI, osteosarcoma resembles chondrosarcoma, presenting as a heterogeneously enhancing, intermediate T1 and high T2 signal mass (Figure 11). However, on CT or radiographs, osteosarcoma usually does not show typical ring-and-arc or whorl shape calcifications.

#### 4.7.6. Metastatic Disease

Most metastases involve the mandibular body and not the condyle; therefore, metastatic disease to TMJ is rare [12]. Metastatic lesions in the TMJ most commonly originate from the breast, lung, prostate, kidney, and thyroid [50,51]. Metastases with different origins, such as melanoma, pancreatic, hepatocellular, and rectal cancer, have been reported in the literature [52,53,54]. The signs and symptoms of TMJ metastatic disease are nonspecific and similar to other TMDs, including pain, restricted motion, clicking, trismus, and malocclusion [54,55,56].

On CT, TMJ metastases present as lytic and destructive bone lesions, but sclerotic or mixed lesions can also be seen with lesions from prostate, lung, and breast origin [57]. TMJ metastases can present with a soft tissue mass with adjacent bone erosion. On MRI, T2 hyperintense and enhancing lesions with adjacent marrow edema may be seen.

Other malignant processes such as lymphoma, multiple myeloma, and malignant synovial sarcoma of the TMJ have been reported [58].

## 5. Summary

Internal derangement is the most common TMJ pathology which can be best evaluated by MRI. The key imaging finding on MR is disc displacement with or without recapture. Other important but less frequent TMJ pathologies are osteoarthritis, idiopathic condylar resorption, inflammatory arthropathies, trauma-related conditions, and tumor and tumor-like lesions.

## Figures and Tables

**Figure 1 diagnostics-12-01006-f001:**
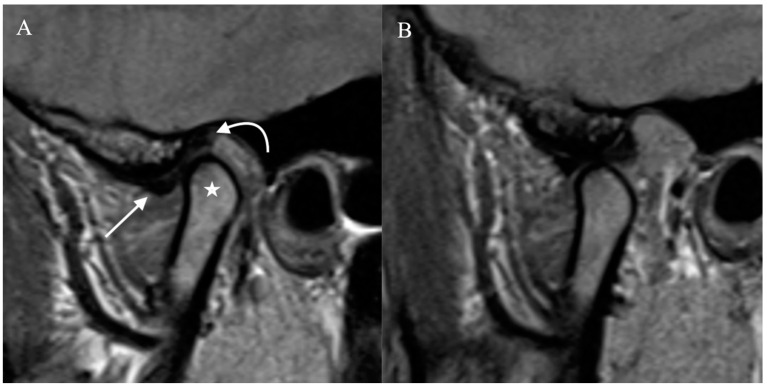
Normal Temporomandibular joint MRI. Proton density sagittal image of the TMJ in closed mouth (**A**) position shows normal location and bow-tie appearance of the articular disc with anterior (straight arrow) and posterior bands (curved arrow). The mandibular condyle (star) is in an anatomic location within the mandibular fossa. On open mouth images (**B**), normal condylar rotation and anterior translation are noted.

**Figure 2 diagnostics-12-01006-f002:**
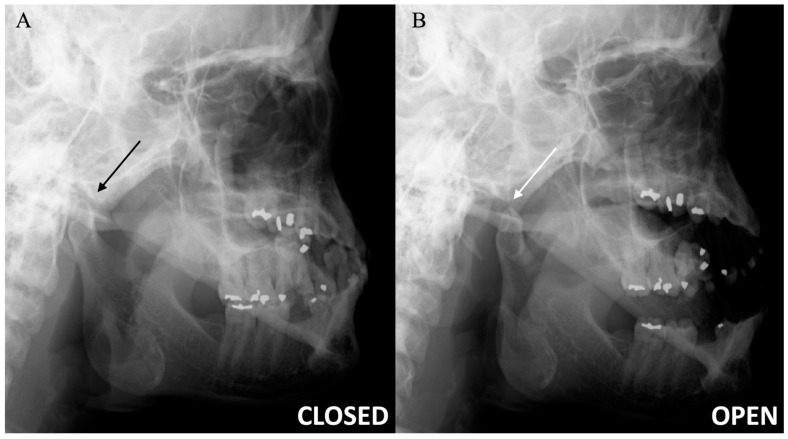
Temporomandibular joint radiographs. Axiolateral TMJ views in the closed mouth (**A**) and open mouth (**B**) positions demonstrate the normal location of the mandibular condyle within the mandibular fossa (black arrow) and normal condylar rotation and anterior translation (white arrow), respectively.

**Figure 3 diagnostics-12-01006-f003:**
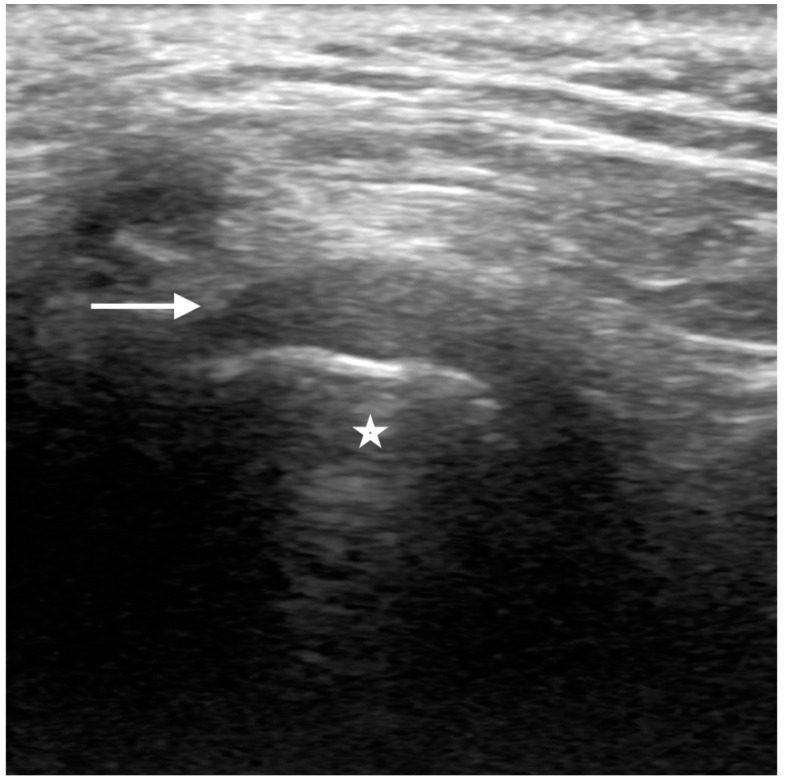
Normal ultrasound of the articular disc. Sonographic images of the TMJ with the probe longitudinal to the articular disc on closed mouth views demonstrate the normal hypoechoic appearance of the mandibular condyle (star), with a rim of the hyperechoic cortex. The articular disc (straight arrow) demonstrates the normal inverted c-shaped morphology and hypoechogenicity, situated just superior to the condylar cortex.

**Figure 4 diagnostics-12-01006-f004:**
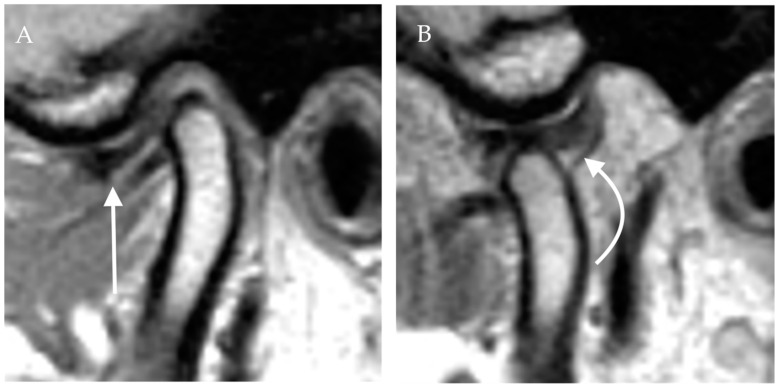
Anterior disc displacement with recapture. Proton density sagittal image of the TMJ in closed mouth (**A**) position shows anterior displacement of the articular disc with otherwise normal bow-tie appearance (straight arrow). The mandibular condyle is situated in an anatomic location within the mandibular fossa. On open mouth images (**B**), normal condylar rotation and anterior translation are noted. Note the recapture of the articular disc, which is now in the normal position (curved arrow). The bow-tie appearance of the articular disc is preserved.

**Figure 5 diagnostics-12-01006-f005:**
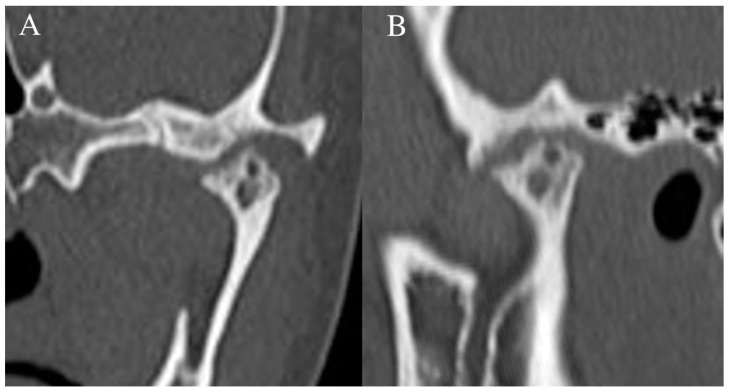
Idiopathic condylar resorption. Sagittal (**A**) and coronal (**B**) images through the TMJ in bone windows demonstrate extensive loss of the mandibular condyle and additional erosive changes of the underlying mandibular fossa and articular eminence.

**Figure 6 diagnostics-12-01006-f006:**
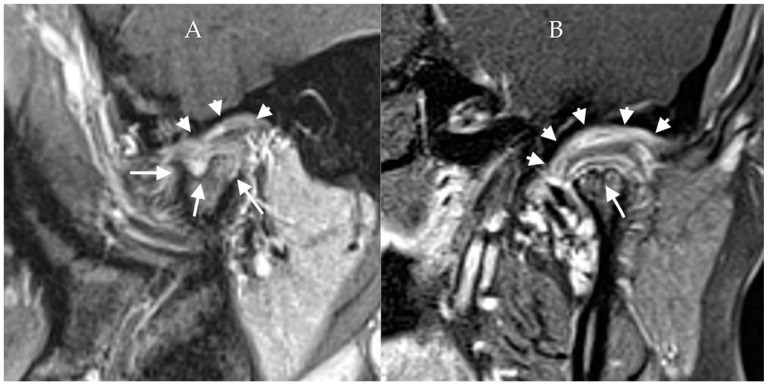
Juvenile idiopathic arthritis. Post-contrast T1W fat-saturated sagittal (**A**) and coronal (**B**) images of the TMJ demonstrate flattening and irregularity of the condyle with erosions (long tail arrows), resulting in an irregular foreshortened appearance from chronic inflammation. Joint effusion is noted with surrounding synovial enhancement (short tail arrows), consistent with an acute JIA flair.

**Figure 7 diagnostics-12-01006-f007:**
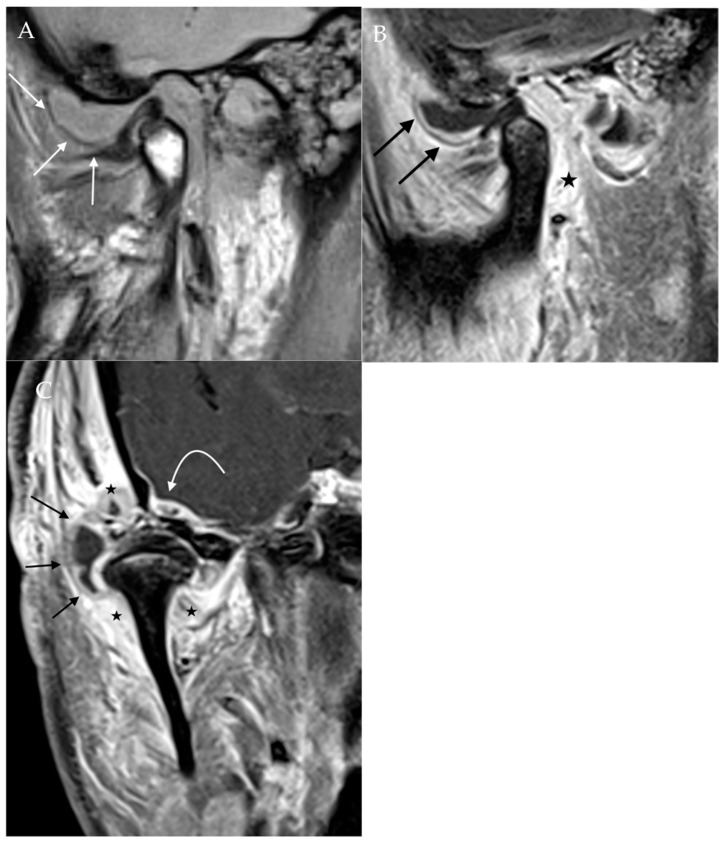
Septic arthritis. Proton density sagittal image of the TMJ (**A**) demonstrates a large joint effusion expanding the space between the articular disc and articular tubercle (straight white arrow). Post-contrast T1Wfat-saturated sagittal (**B**) and coronal (**C**) images demonstrate extensive surrounding synovial enhancement (black arrows) and soft tissue enhancement (star). Also noted is enhancement along the dura of the right temporal lobe (curved arrow), indicating the intracranial extension of infection.

**Figure 8 diagnostics-12-01006-f008:**
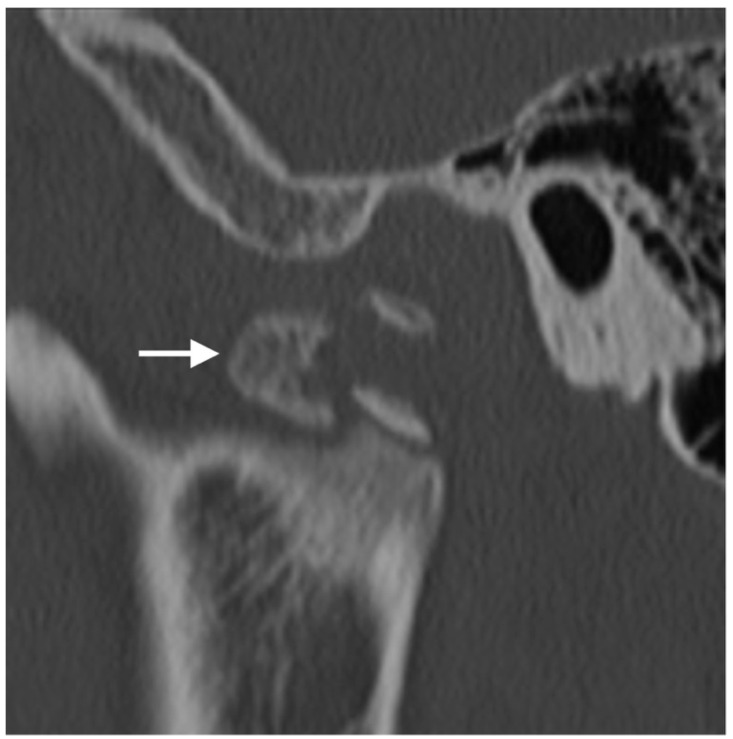
Trauma. Sagittal CT image of the TMJ demonstrates comminuted fracture of the condylar neck with a displacement of the fracture fragments. Mild sclerosis around the fracture lines suggests a component of interval healing. The tip of the mandibular condyle (arrow) is displaced antero-inferiorly.

**Figure 9 diagnostics-12-01006-f009:**
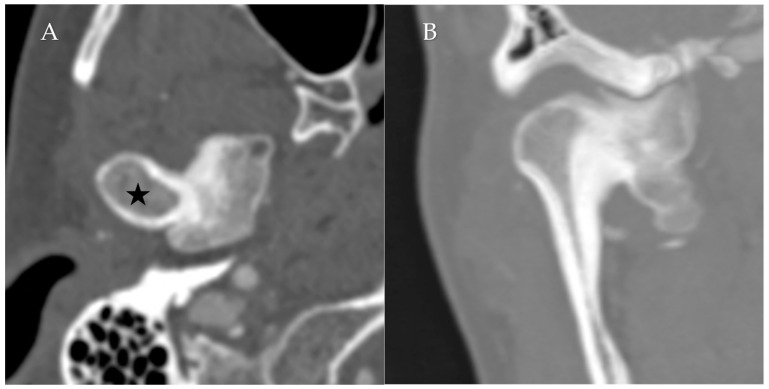
Condylar Osteochondroma. Axial (**A**) CT in bone windows and reformatted coronal (**B**) image demonstrates a prominent bony mass arising along the medial aspect of the mandibular condyle (star), with a well-corticated appearance and no aggressive features.

**Figure 10 diagnostics-12-01006-f010:**
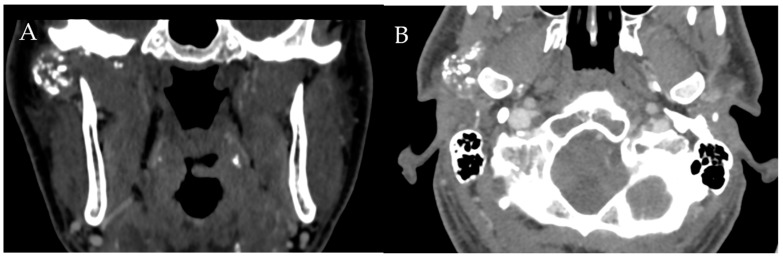
Synovial Chondromatosis. Coronal (**A**) and axial (**B**) CT shows multiple calcified loose bodies in the right TMJ consistent with synovial chondromatosis.

**Figure 11 diagnostics-12-01006-f011:**
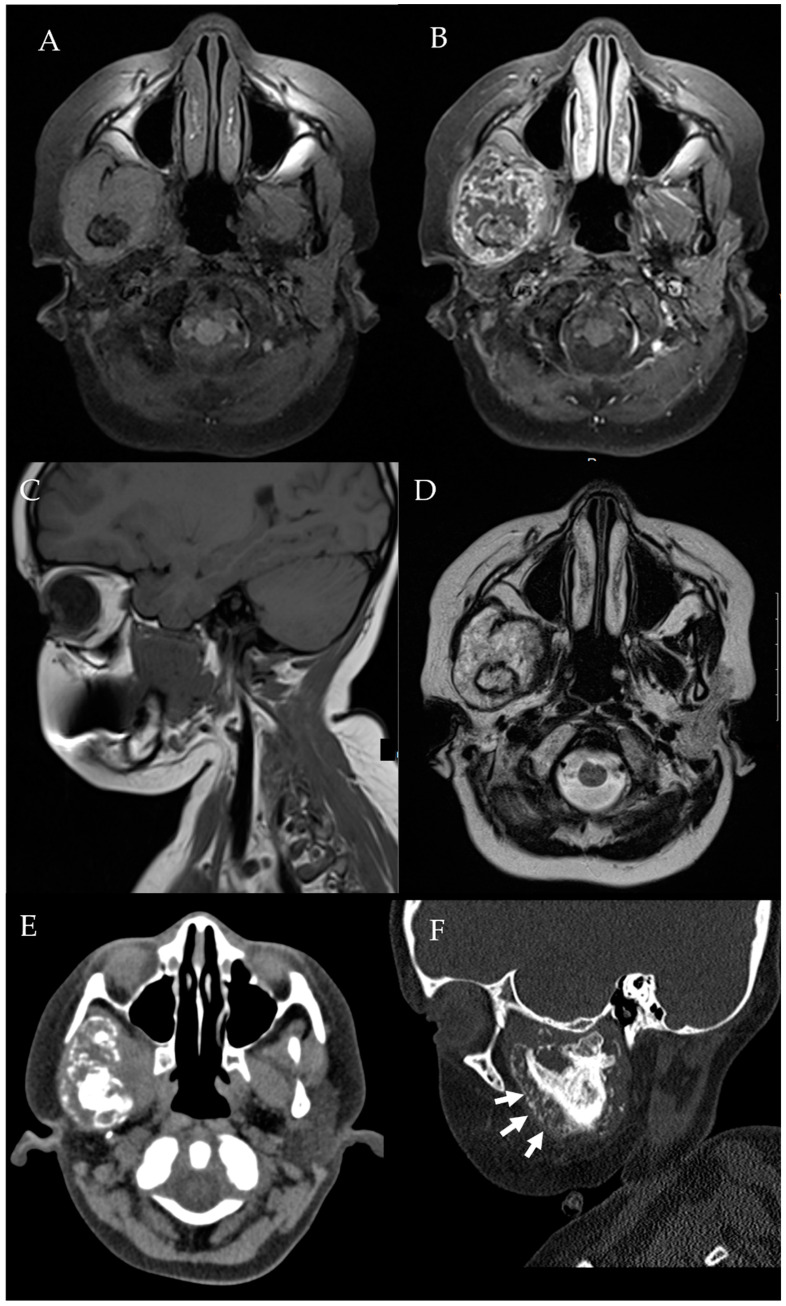
Condylar chondroblastic osteosarcoma. Axial pre-contrast (**A**) and post-contrast T1W (**B**), sagittal pre-contrast T1W (**C**), and axial T2W (**D**) images demonstrate a heterogeneously enhancing and T2-hyperintense right condylar mass in a nine-year-old patient. Axial (**E**) and sagittal (**F**) CT images show a soft tissue mass with mineralized matrix and aggressive periosteal reaction (small arrows) at the TMJ. Biopsy confirmed chondroblastic osteosarcoma.

## Data Availability

Not applicable.

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
