# Peer review of "Imaging of the Temporomandibular Joint"

_diagnostics, 2022, doi:10.3390/diagnostics12041006_

Round 1
Reviewer 1 Report
At first glimpse this review might be considered a student textbook knowledge because as a narrative review does not bring much new into body of evidence about TMJ diagnostics and radiology. And as such it should be published in non-peer reviewed journal. On the other hand, article is very well and comprehesively written yet emphasizes some radiological findings on rare TMJ diseases. I'd consider publishing it after some major changes. My remarks below:
Abstract - this section should be structurized according to MDPI policy
Keywords: adding wisely a few more according to MeSH would improve availaibilty through search engines, which will lead to improved citation odds eventually
L30-53 - these statements are missing relevant citation(s)
L47-L53 - these sentences are oversimplified - it is clearly visible now that authors are radiologists who are usually not very much into TMJ biomechanics, TMDs and Orofacial Pain - these statements are way too obsolete in early 2022. I understand the good intentions, however my advice is that authors should start with reading some essential textbooks first e.g. Tamimi, Okeson, Sharav&Benoliel to mention just these few prior to writing an article on such vast topic.
Fig 1 - image quality might have been better
L56-57 - 'anterior and posterior articular disc' - this is definitely not as it stands in anatomy texbooks - please re-formulate
Fig 2 - Normal TMJ radiograph? I have never seen such technique before - did authors mean Schiller technique or other? Please specify.
L98-98 - ultrasonographys' major flaw is not only being operator sensitive, yet has restricted access to deeper structures - this should be emphasized here either as a serious limitation.
L110 - there are much newer references available as a replacement for [8] - I suggest https://pubmed.ncbi.nlm.nih.gov/33101543/ and https://pubmed.ncbi.nlm.nih.gov/30026884/
L148-154 - these statements are missing relevant citation(s)
L157-158 - not so poorly, I suggest incorporating and citing at least https://www.mdpi.com/2077-0383/11/4/952 and https://www.mdpi.com/2077-0383/9/4/1159
L170 - 'muscle spasms' is not a valid term, please re-formulate
L174-175 - ' Interestingly, articular disc displacement is not a significant finding in patients with RA' - there is a conflicting evidence about this, please add a few more relevant citation(s)
L271 - did authors mean insertion?
Authors should write a short summary of their findings at the end of article.
Reviewer 2 Report
This is an excellent paper - I have only minor comments.
In the Introduction, is the $4 billion annual cost worldwide or just in the USA ?
Fig 1 B : I suggest adding : A "bow-tie" appearance is usually seen.
Fig 4 B : add : and the "bow-tie" appearance is still seen. Note that OMFS surgeons ( like me ) commonly use this term to describe the appearance of the undisplaced disc in the open-mouth position.
Do the authors have an image of Idiopathic Condylar Resorption ? It would enhance the paper.
This will be a useful addition to the literature.
